# Seasonal Variation of Melatonin Concentration and mRNA Expression of Melatonin-Related Genes in Developing Ovarian Follicles of Mares Kept under Natural Photoperiods in the Southern Hemisphere

**DOI:** 10.3390/ani13061063

**Published:** 2023-03-15

**Authors:** Lia Alencar Coelho, Luciano Andrade Silva, Ana Paula Reway, Daniella Do Carmo Buonfiglio, Jéssica Andrade-Silva, Patrícia Rodrigues Lourenço Gomes, José Cipolla-Neto

**Affiliations:** 1Department of Veterinary Medicine, Faculty of Animal Science and Food Engineering (FZEA), University of São Paulo, Pirassununga 13635-900, Brazil; 2Department of Physiology and Biophysics, Institute of Biomedical Sciences (ICB), University of São Paulo, São Paulo 05508-000, Brazilcipolla@icb.usp.br (J.C.-N.); 3Department of Pharmacology, Toxicology and Neuroscience Institute, Morehouse School of Medicine, Atlanta, GA 30310, USA

**Keywords:** *Asmt*, *Aanat*, follicular fluid, gonadotropin receptors, mares, melatonin receptors

## Abstract

**Simple Summary:**

Reproductive activity in mares shows a seasonal pattern that is associated with an increasing photoperiod with a pronounced incidence of ovulations during spring and summer. The photoperiodic control of reproductive activity is mediated by melatonin, secreted by the pineal gland, which plays an inhibitory role in ovulatory activity via the hypothalamic-pituitary axis. Considering that there is little information on the direct effect of melatonin on the development of equine ovarian follicles, we studied the seasonal variations in mRNA expression of gonadotropin receptors (*Fshr* and *Lhr*), melatonin receptors (*Mt1* and *Mt2*), melatonin-synthetizing enzymes (*Asmt* and *Aanat*) and melatonin concentration in developing follicles (small—<20 mm, medium—20 to 35 mm and large—>35 mm) from five mares raised in natural photoperiods. There was an increased mRNA expression of gonadotropin receptors and melatonin-related genes and an increase of melatonin levels in developing follicles during the spring/summer seasons. The total number of large follicles (potential ovulatory follicles) was significantly higher during the spring/summer seasons. Our results demonstrate that melatonin upregulates the mRNA expression of melatonin receptors and melatonin-forming enzymes in mare developing follicles during reproductive seasons.

**Abstract:**

This study investigated the seasonal variations in mRNA expression of FSH (*Fshr*), LH (*Lhr*) receptors, melatonin (*Mt1* and *Mt2*) receptors, melatonin-synthetizing enzymes (*Asmt* and *Aanat*) and melatonin concentration in developing follicles from mares raised in natural photoperiods. For one year, ultrasonographic follicular aspiration procedures were performed monthly, and small (<20 mm), medium (20 to 35 mm) and large (>35 mm) follicles were recovered from five mares. One day before monthly sample collections, an exploratory ultrasonography conducted to record the number and the size of all follicles larger than 15 mm. The total number of large follicles were higher during the spring/summer (8.2 ± 1.9) than during autumn/winter (3.0 ± 0.5). Compared to autumn/winter seasons, there was an increase of *Fshr* and *Aanat* mRNA expressions in small, medium and large follicles, an increase of *Lhr* and *Asmt* mRNA expressions in medium and large follicles and an increase of *Mt1* and *Mt2* mRNA expressions in small and large follicles during spring/summer. The melatonin levels in follicular fluid were also higher during the spring/summer seasons. The present data show that melatonin locally upregulates the mRNA expression of *Mt1* and *Mt2* receptors and melatonin-forming enzymes in mare developing follicles during reproductive seasons.

## 1. Introduction

The incidences of ovulatory activity in mares are dependent on the time of year. During the winter, the incidence of ovulation is minimal or absent, increasing during the spring, it becomes maximum in summer, and decreases during the autumn [1]. Although the incidence of ovulations presents a seasonal pattern, estrus behavior in mares can be expressed even during the anovulatory period [1,2]. The seasonal pattern of reproductive activity comes from an endogenous circannual rhythm, which is determined by several environmental factors including the photoperiod [3]. The beginning of the reproductive activity in mares is related to the increase in daylight hours, and an additional lighting exposure during short-day seasons can anticipate the start of the breeding season [3,4]. Several studies have shown that the neuroendocrine control of seasonal reproduction in mares is mediated by melatonin secreted by pineal gland [5,6]. The administration of melatonin in subcutaneous implants to pony mares during the summer produces a significant reduction of hypothalamic GnRH [7] and alters the secretion of LH [8] according to the ovarian status (intact or ovariectomized). In addition, exposing mares to constant light decreases the concentration plasma melatonin and increases hypothalamic GnRH [9], suggesting a regulatory role for melatonin at the central level.

In addition to the effects of melatonin on reproduction being mediated mainly through its action in hypothalamic and pituitary regions [10,11], numerous studies have shown that melatonin may act directly on the regulation of ovarian function [12,13,14,15]. During the ovarian follicle development, melatonin may act in the estrogen and progesterone productions [16,17,18] by regulating the steroidogenic gene expressions [16,19] including the mRNA expression of LH receptor [17,20] through the MT1 and MT2 receptors present in granulosa-luteal cells [17,21,22]. Melatonin also protects the oocyte from oxidative stress [23] improving oocyte’s quality whose effects could be attributed to its free radicals scavenging action [24]. The concentration levels of melatonin in follicular fluid [25], probably synthetized by the ovary [26,27,28], varied according to the human [14] and porcine [29] follicle sizes, and circadian and seasonal changes in melatonin concentration were also detected in human preovulatory follicular fluid [30]. In general, melatonin plays a positive effect on oocyte maturation, fertilization, and embryo development as well in the direct regulation of steroidogenesis [12,13,14,15,16,18,19,20,21,22,23,27,28,29] during the follicle development. Because of these physiological characteristics, melatonin plays an important role in human assisted reproductive technologies (ART) by improving the clinical outcomes of IVF-embryo transfer [13,31]. Like in humans, ART procedures were also used in horses and some similarities in these technologies have been shown between women and mares [32].

Considering that there is evidence of a direct effect of melatonin on the equine ovary showing no photoperiodic differences in mRNA expression of *Mt1* receptors in developing follicles [17], we studied the seasonal variations in mRNA expression of FSH (*Fshr*), LH (*Lhr*), melatonin (*Mt1* and *Mt2*) receptors, melatonin-synthetizing enzymes (*Asmt* and *Aanat*) and melatonin concentration in developing follicles from mares raised in natural photoperiods.

## 2. Materials and Methods

### 2.1. Mares

Five fertile crossbred mares from the Laboratory of Theriogenology Dr. O. J. Ginther of Faculty of Engineering and Animal Science, University of São Paulo (aged 6 to 14 years and weighing 300–400 kg) were used in this study. All animals were clinically healthy with no evidence of diseases or reproductive disorders. Mares were kept under natural photoperiods in Southeast Brazil (21°59 Southern latitude and 47°26 Western longitude from Greenwich and 634 m altitude). The mares were kept exclusively on pasture of Mombaça grass (*Panicum maximum Jacq. cv. Mombaça*), with access to water and trace-mineralized salt ad libitum. They also received individual and daily protein-energy maintenance supplementation.

### 2.2. Aspiration Procedure and Follicular Cells Recovery

The aspiration procedures were carried out every month for one year. One day before the follicular aspiration, an exploratory ultrasonography was performed for ovarian assessment when the number and the size of all follicles larger than 15 mm and luteal development were recorded. Immediately before the aspiration procedures, the mares were sedated with Butorphanol (0.01 mg/kg) and Romifidine (0.04 mg/kg). An ultrasound machine with a multifrequency linear transducer (5–8 MHz) was used for follicular aspiration procedures [33]. Each follicle was aspirated with a 12-gauge double lumen needle (WTA^®^, Cravinhos, SP, Brazil) attached to a vacuum pump whose pressure varied between 150 mmHg and 250 mmHg according to the size of the follicles. The probe protected with a sanitary plastic condom was inserted transvaginally for follicular aspiration, and the ovaries were manipulated transrectally and placed next to the ultrasound probe for visualization of the follicles. Data were collected during the light phase of the day, between 8:00 a.m. and 11:00 a.m.

The aspirated follicles were collected in tubes containing Tissue Culture Medium 199 with HEPES (Gibco^®^/12350-039, Invitrogen, Thermo Fisher Scientific, Waltham, MA, USA) [20]. To avoid any individual differences, the follicular (granulosa and theca cells) cells were isolated from a pool of follicles from different sizes which were categorized as small (<20 mm), medium (20 to 35 mm) and large (>35 mm). After the collection, the separated samples were put into 1000 µL of lysis solution (TRIzol^®^, 15596-018, Invitrogen, Thermo Fisher Scientific, Waltham, MA, USA) for RNA isolation and stored at −80 °C. After centrifugation at 4 °C for 10 min, the follicular fluid samples from different sizes of follicles were also stored at −80 °C.

### 2.3. mRNA Extraction and qRT-PCR

For RNA extraction, the TRIzol^®^ RNA isolation protocol was used [22]. The aqueous phase of the thawed sample, previously diluted in TRIzol^®^ reagent, was separated with 200 µL of chloroform. The RNA precipitation and the washing of RNA pellet were carried out using 500 µL of isopropanol alcohol and 1000 µL of 75% ethanol, respectively. Then, the RNA pellet was dissolved in DEPC treated water. The RNA quantification was conducted at 260 mm on Nanodrop^TM^ 1000 equipment (Thermo Fisher Scientific, Waltham, MA, USA).

The first-strand cDNA was produced from 1 µg of total RNA pellet diluted in RNase-free water using reverse transcriptase (Superscript III, 18080-044, Invitrogen, Thermo Fisher Scientific, Waltham, MA, USA) and random primers (48190-011, Invitrogen, Thermo Fisher Scientific, Waltham, MA, USA). The Quantitative Reverse Transcriptase Polymerase Chain Reaction (qRT-PCR) was performed on QuantStudio 6 Flex Real-Time PCR equipment (Applied Biosystems, Inc., Foster City, CA, USA) using the relative quantification analyses (2^−ΔΔCT^ method) [34] and reported as arbitrary units. The PCR efficiency of the target and reference genes was 95%. All samples were analyzed in duplicate, and *Gapdh* was used as a reference gene [35]. The accession number of the genes and the primer sequences are listed in Table 1.

### 2.4. Melatonin Concentration in Follicular Fluid

The concentrations of melatonin (pg/mL) in follicular fluid samples were collected during the light phase of day and measured using a commercial melatonin ELISA kit (IBL International, Hamburg, Germany) according to the manufacturer’s instructions. The samples were assayed in duplicate, and the sensitivity of the assay was 0.5 pg/mL.

### 2.5. Statistical Analysis

Data were expressed as mean ± SEM, calculated from at least five replications. Seasonal (spring/summer vs. autumn/winter) effects on the number of follicles and the mRNA expression in each class of follicles were analyzed using the Welch’s *t*-test and the non-parametric Mann–Whitney test, respectively. Seasonal effects on melatonin concentrations in follicular fluid (FF) were analyzed using the two-way ANOVA followed by Bonferroni’s post-test. A correlation analysis between melatonin levels in FF and mRNA expression of melatonin-related genes was also performed. All analyses were performed using GraphPad Prism (GraphPad Software version 9.40; San Diego, CA, USA).

## 3. Results

### 3.1. Annual Distribution of the Number of Developing Ovarian Follicles and Seasonal Variation of mRNA Expression of FSH (Fshr) and LH (Lhr) Receptors in Ovarian Follicles of Different Sizes

The ovaries of five mares were monitored monthly by ultrasonography for one year. The total number of large follicles were significantly higher during spring/summer than during autumn/winter (Table 2). Considering the mRNA expression of *Fshr* and *Lhr* in follicles from different sizes (Figure 1), the small, medium and large follicles presented an increase of *Fshr* mRNA expression during the spring/summer seasons. The *Lhr* mRNA expression in medium and large follicles was also increased during spring/summer seasons.

### 3.2. Seasonal Variation of mRNA Expression of Melatonin-Related Genes in Ovarian Follicles of Different Sizes

The mRNA expression of melatonin receptors (*Mt1* and *Mt2*) in ovarian follicles is presented in Figure 2. The expression of both receptors in small and large follicles was higher in spring/summer seasons than in autumn/winter seasons. No seasonal variation of *Mt1* and *Mt2* mRNA expressions was observed in medium follicles. Figure 3 shows the mRNA expression of melatonin synthetizing enzymes (*Amst* and *Aanat*). A significant increase of *Asmt* and *Aanat* mRNA expressions was observed in medium and large follicles during the spring/summer. In small follicles, the mRNA expression of *Aanat* was also increased during spring/summer seasons, but no seasonal difference of *Asmt* mRNA expression was observed.

### 3.3. Seasonal Variation on Melatonin Concentration in Follicular Fluid from Developing Follicles

Considering seasonal melatonin concentration differences in developing follicles (Figure 4), melatonin levels in medium and large follicles were significantly higher during the spring/summer seasons. During the autumn/winter seasons, no differences in melatonin concentration were observed in small follicles. The correlation analysis (Table 3) revealed a significant positive correlation between seasonal melatonin levels in follicular fluid and the seasonal mRNA expression of melatonin forming enzymes (*Asmt* and *Aanat*) in medium and large follicles. In large follicles, seasonal melatonin concentrations in follicular fluid were also positively correlated with seasonal variation in mRNA expression of melatonin receptors (*Mt1* and *Mt2*). No significant correlation was observed in small follicles.

## 4. Discussion

The present study investigated the seasonal variation of melatonin concentration and mRNA expression of melatonin-related genes in developing follicles of mares living under natural photoperiods in Southeast Brazil. First, we verified the annual follicular activity by monitoring the ovaries one day before the monthly collection of the samples. The ovarian follicles were classified according to the follicular dynamics that are characterized by the development of an ovulatory and anovulatory major waves and anovulatory minor waves that are mainly regulated by circulating gonadotropin levels [36]. In mares, the emergence of a follicular wave occurs in response to FSH surge which reach the peak when a cohort of follicles attains approximately 13 mm in diameter (small follicles). The declining FSH levels lead to the selection of one largest follicle with 21 to 23 mm in diameter (medium follicles) which deviates in diameter from the smaller follicles (subordinate follicles) that go into regression. After the deviation, the largest follicle continuously grows, becoming a dominant follicle (large follicles) with approximately 35–45 mm in diameter which may ovulate or stop growing and regress, depending on the occurrence of the LH surge [36].

Our research revealed that the number of large follicles was significantly higher during the spring/summer seasons confirming the seasonal follicular activity whose annual changes in photoperiod have been the main environmental factor that synchronizes this seasonality in temperate, subtropical and tropical environments [3,4,5,37]. It has been reported that the mare follicular activity decreases during the winter, and the follicular waves during the non-breeding season are characterized by the absence of large follicles and unaltered presence of small and medium follicles [38]. We also analyzed the mRNA expression of FSH and LH receptors, which reflects the action of these gonadotropins in developing follicles [39]. The present study revealed that, during spring/summer seasons, there was a significant increase of *Fshr* mRNA expression in small, medium and large follicles and an increase of *Lhr* mRNA expression in medium and large follicles. The development of follicular waves has been associated with FSH surge for major and minor waves. In ovulatory, transitional and anovulatory seasons, FSH concentration has been more important for follicular growth before deviation while LH plays an important role during deviation and in stimulating the development of dominant follicles [36,38]. Moreover, it has been shown that no seasonal differences were observed in circulating FSH associated with follicular activity, and significant LH requirements are determinant for the growth of large follicles during the ovulatory and anovulatory seasons [38]. However, large follicles have shown a higher responsiveness to LH during the ovulatory season (spring/summer) than during the anovulatory season (autumn/winter) as indicated by high levels of *Lhr* mRNA expression in large follicles during the ovulatory season [40]. Our results suggest that, during the spring and summer seasons, small, medium and large follicles have an increased responsiveness to circulating FSH, and medium and large follicles have an increased responsiveness to LH levels.

Our findings also showed that the mRNA expression of melatonin-related genes in developing follicles and melatonin concentration in follicular fluid varied seasonally. The mRNA expression of *Mt1* and *Mt2* receptors in small and large follicles was significantly elevated during the spring/summer seasons. In the same way, we also observed an increase of mRNA expression of *Asmt* and *Aanat* and an increase of melatonin levels in developing follicles during the spring/summer seasons. These results suggest a putative direct effect of local producing melatonin on the ovarian follicle development independent of the seasonal effects of melatonin on the photoperiodic control of reproductive activity via the hypothalamic-pituitary axis. As previously mentioned, melatonin is an important signal in the seasonal control of reproductive activity in mares [6,7,8,9,10]. The breeding season period is associated with increasing daylength and takes place during the spring and summer months when the duration of the nocturnal melatonin secretion reflects the length of scotophase [11,41]. In fact, it has been demonstrated that pinealectomy and melatonin treatment at the timing of onset of nocturnal melatonin secretion blocked the ability of mares in responding to stimulatory long photoperiods [6,7,9,42]. On the contrary, our results showed an increase of the mRNA expression of the melatonin related genes (*Mt1*, *Mt2*, *Asmt* and *Aanat*) and an increase of melatonin concentration in developing follicles during the spring and summer months, suggesting a direct effect of local synthetized melatonin on follicle development in the same period of ovulatory activity when melatonin signal is reduced. In addition to confirming the previously reported direct effect of melatonin on ovarian function in mares [17], the present findings also suggest that melatonin is acting locally in all steps of follicular wave development. Apparently, melatonin action has been more pronounced in dominant follicles which have the potential to ovulate. The increase of mRNA expression of melatonin-synthetizing enzymes and melatonin levels in large follicles revealed a local and beneficial role of melatonin in the final follicular growth during the ovulatory season. One of the most important effects of melatonin on ovarian follicle growth is to protect the oocyte and follicular cells from intrafollicular oxidative stress by reducing reactive oxygen species (ROS) and acting as a potent free radical scavenger during the ovulation process [14,15,23] since ovulation is like a local inflammatory response whose ROS are generated [43]. Our results are compatible with the premise that melatonin in follicular fluid was locally secreted by oocytes and/or follicular cells [26,27,28], and the elevated melatonin levels during the reproductive season play a beneficial role in follicular growth and protection. It has been demonstrated that high concentrations of melatonin in follicular fluid were positively correlated with the quantity of good-quality oocytes, oocytes fertilized and good-quality embryos [44,45], critical conditions for the success of assisted reproductive technologies [13].

The present research shows that melatonin has the potential to be used in equine ART programs [46] by acting as an antioxidant agent for in vitro maturation cultures [47]. Indeed, the role of melatonin as an antioxidant in human ART programs has recently been well documented [48]. Melatonin could be used as a supplement in culture media improving oocyte maturation and fertilization and embryonic development [13,23,31,49]. Oral administration of melatonin to patients in human ART programs was able to improve fertilization rates and the quality of oocytes and embryos [23,50]. Additionally, these results could be also considered important as a basis for further studies which focus on human ART programs since there are similarities in the dynamics of follicular growth between the two species [32,51].

## 5. Conclusions

The current findings demonstrate that melatonin, probably from peripheral ovarian melatonin synthesis, upregulates the mRNA expression of melatonin receptors and melatonin-forming enzymes in mare developing follicles during the reproductive season.

## Figures and Tables

**Figure 1 animals-13-01063-f001:**
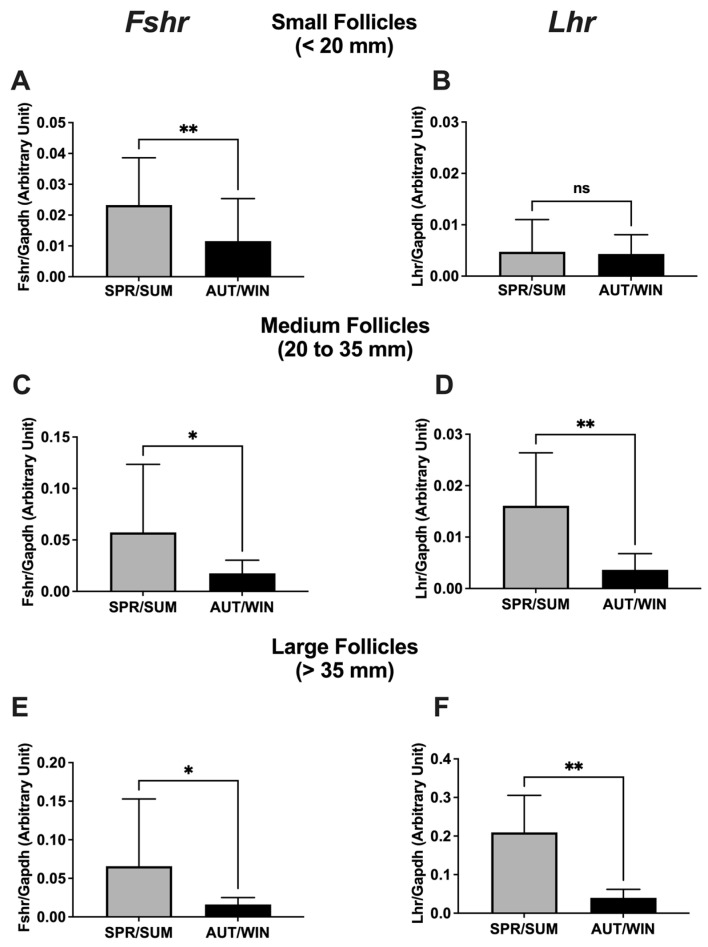
Seasonal distribution of the mRNA expression of *Fshr* (**A**,**C**,**E**) and *Lhr* (**B**,**D**,**F**) genes in mare ovarian follicles. SPR/SUM: spring and summer; AUT/WIN: autumn and winter. Non-parametric Mann–Whitney test was used to determine significant differences (plotted as mean ± SEM). Asterisks represent season differences (* *p* < 0.05; ** *p* < 0.01). ns: non-significant. Reference gene: *Gapdh*.

**Figure 2 animals-13-01063-f002:**
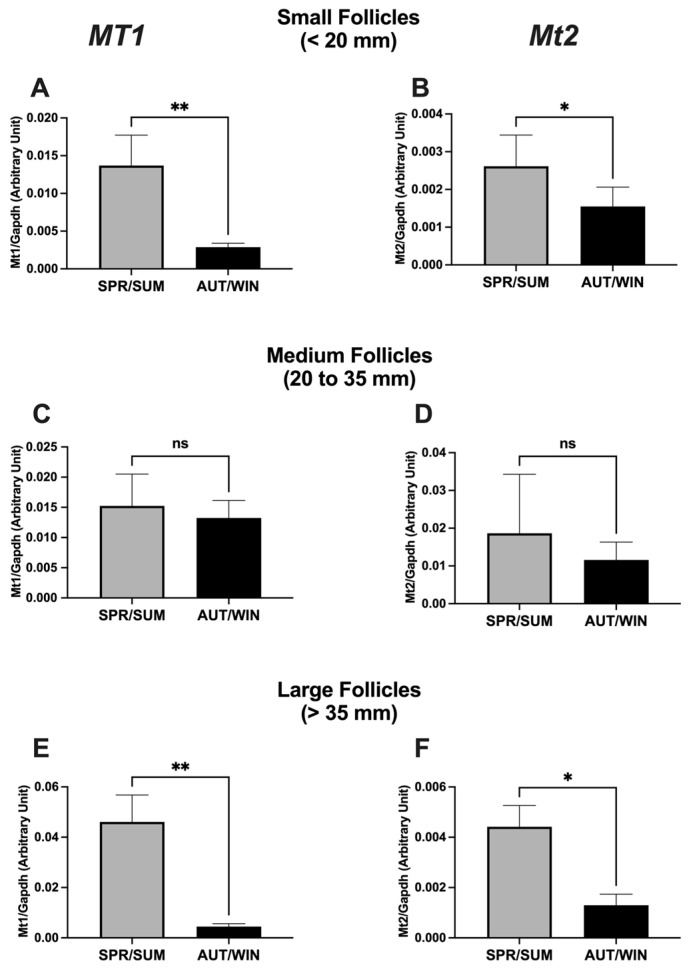
Seasonal distribution of mRNA expression of *Mt1* (**A**,**C**,**E**) and *Mt2* (**B**,**D**,**F**) genes in mare ovarian follicles. SPR/SUM: spring and summer; AUT/WIN: autumn and winter. Non-parametric Mann–Whitney test was used to determine significant differences (plotted as mean ± SEM). Asterisks represent season differences (* *p* < 0.05; ** *p* < 0.01). ns: non-significant. Reference gene: *Gapdh*.

**Figure 3 animals-13-01063-f003:**
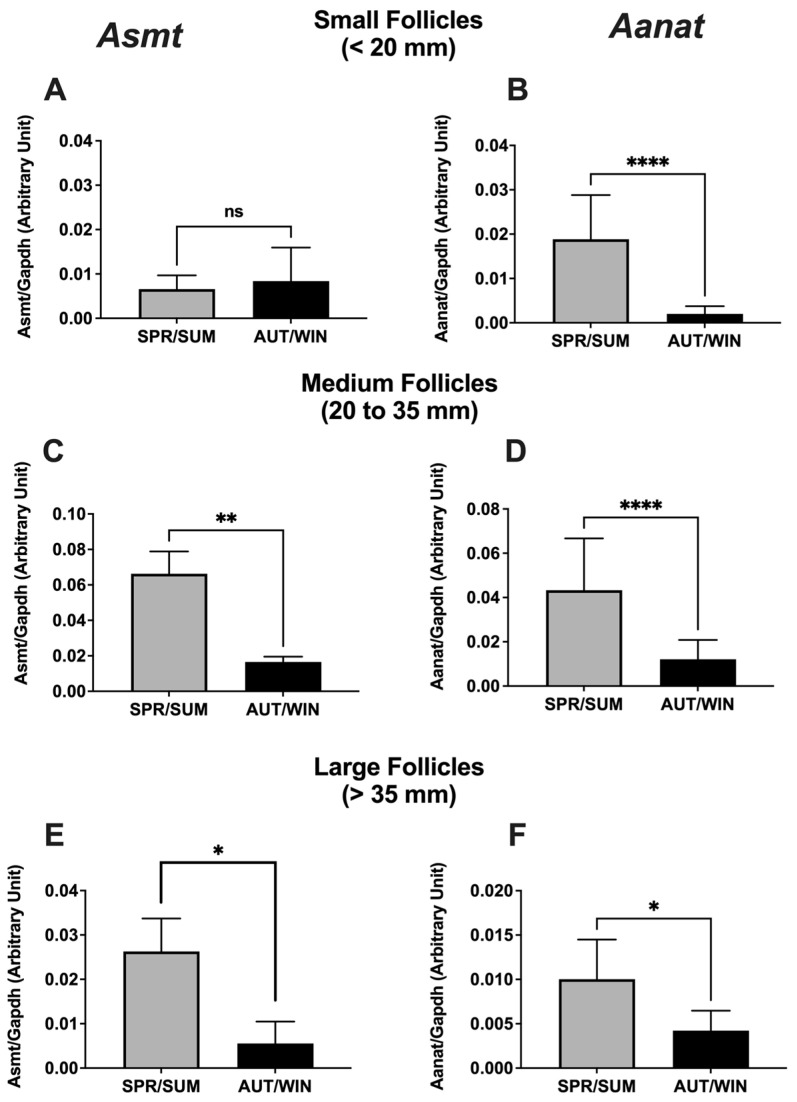
Seasonal distribution of mRNA expression of *Asmt* (**A**,**C**,**E**) and *Aanat* (**B**,**D**,**F**) genes in mare ovarian follicles. SPR/SUM: spring and summer; AUT/WIN: autumn and winter. Non-parametric Mann–Whitney test was used to determine significant differences (plotted as mean ± SEM). Asterisks represent season differences (* *p* < 0.05; ** *p* < 0.01, **** *p* < 0.001). ns: non-significant. Reference gene: *Gapdh*.

**Figure 4 animals-13-01063-f004:**
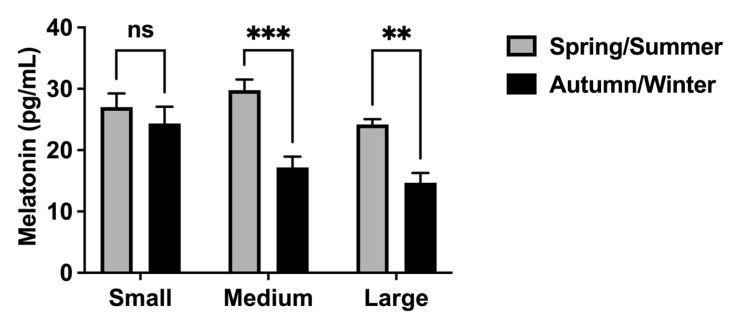
Seasonal melatonin concentrations (pg/mL) in mare follicular fluid from different sizes of ovarian follicles. Small: <20 mm; Medium: 20 to 35 mm; Large: >35 mm. Two-way ANOVA with Bonferroni’s multiple comparisons post-test was used to determine significant differences (plotted as mean ± SEM). Asterisks represent seasonal differences (** *p* < 0.01, *** *p* < 0.001). ns: non-significant.

**Table 1 animals-13-01063-t001:** List of qRT-PCR primers used in this study.

Target and Reference Genes	Accession Number *	Primer Sequence 5′–3′	Base Pairs (bp)
*Mt1*	XM_001490171.2	F: 5′-GCTGAGGAACGAGGAAA-3′	109
		R: 5′-GATGTCAGCACCAAGGGTAT-3′	
*Mt2*	XM_001917051.1	F: 5′-CCGGAACCCAGGTAATTTGT-3′	108
		R: 5′-GCCCAGCCATCATGGAAGA-3′	
*Asmt*	XM_023633485.1	F: 5′-TCCGATTCTGTGAAGGCGAT-3′	103
		R: 5′-TGGGCACATTTCTCATCCGT-3′	
*Aanat*	XM_023651967.1	F: 5′-CCCCACAGTCCTGGTGCTC-3′	105
		R: 5′-CTCTGTGTGGACATCCTGGC-3′	
*Fshr*	NM_001164013.1	F: 5′-GAACCCAACTAGATGAGCTGAAT-3	81
		R: 5′-CAGAGGCTCCCTGGAAAACA-3′	
*Lhr*	AY394748.1	F: 5′-GGAACACTTTATTCTGCCAGC-3′	94
		R: 5′– GGAGCACATCGGAGTGTCTT-3′	
*Gapdh*	NM_001163856.1	F: 5′-GTCGGAGTAAACGGATTTGGC-3′	91
		R: 5′-TGAAGGGGTCATTGATGGCG-3′	

* Accession Number is provided by the National Center for Biotechnology. F: Forward, R: Reverse.

**Table 2 animals-13-01063-t002:** Seasonal distribution of the number of follicles (Means ± SEM) evaluated one day before the follicular cell recovery.

Seasons	Number of Mares ^1^	Total Number of Follicles for Each Mare			Number of Follicles per Mare/Month		
		Small	Medium	Large	Small	Medium	Large
spring/summer	05	61.0 ± 9.9	7.2 ± 1.8	8.2 ± 1.9 *	9.5 ± 0.6	1.1 ± 0.1	1.2 ± 0.1 *
autumn/winter	05	45.4 ± 11.1	6.4 ± 1.7	3.0 ± 0.5	10.9 ± 0.2	1.4 ± 0.2	0.7 ± 0.1

^1^ The same five mares were used throughout the year. Small: <20 mm; Medium: 20 to 35 mm; Large: >35 mm. Asterisks represent significant differences (* *p* < 0.05) between values in the same column.

**Table 3 animals-13-01063-t003:** Correlation between melatonin (Mel) concentration in follicular fluid (FF) and mRNA of melatonin-related genes in developing follicles (small, medium and large).

Mel FF (X)	*Asmt* (Y)		*Aanat* (Y)		*Mt1* (Y)		*Mt2* (Y)	
	Pearson r	*p*-Value	Pearson r	*p*-Value	Pearson r	*p*-Value	Pearson r	*p*-Value
Small	0.4327	0.4668	−0.1666	0.7211	−0.1747	0.6293	0.0052	0.9887
Medium	0.7806	0.0130 *	0.6770	0.0040 **	0.0005	0.9986	−0.1041	0.7013
Large	0.7773	0.0397 *	0.7824	0.0376 *	0.6223	0.0409 *	0.7344	0.0380 *

Small: <20 mm; Medium: 20 to 35 mm; Large: >35 mm. Asterisks represent significant differences (* *p* < 0.05; ** *p* < 0.01).

## Data Availability

These data are available on request from corresponding author.

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
