# Peer review of "Seasonal Variation of Melatonin Concentration and mRNA Expression of Melatonin-Related Genes in Developing Ovarian Follicles of Mares Kept under Natural Photoperiods in the Southern Hemisphere"

_animals, 2023, doi:10.3390/ani13061063_

Round 1

Reviewer 1 Report

This paper seeks to correlate levels of melatonin in follicular fluid across seasons to gene expression in small, medium and large follicles.  This is a novel study in the horse; however, there are several errors in the methodology which must be corrected before this paper is suitable for publication.  

Major

Authors state that they examined follicular mRNA.  From the description, it appears to be the granulosa cells which are assessed, but that needs to be clarified throughout.  Please also discuss whether granulosa cell status was assessed (compact vs expanded) as this will affect gene expression

The data are not appropriately normalized.  In the methodology, you state you use 2^(-deltaCt), but this isn’t a valid methodology, and neither is GeneX/GAPDH as is stated in the y-axis of the graphs.  I would recommend using -deltaCt for graphs and statistics (e.g. -(geneX Ct – GAPDH Ct)). 

Although there appears to be a strong correlation between expression of these genes and melatonin concentrations, this is only correlation and all discussion about melatonin affecting expression of other genes must be removed. 

Authors may want to run correlations between melatonin levels and gene expression levels to improve the manuscript. 

Although outside the scope of this manuscript, it would be interesting to inject melatonin into follicles in the fall to better determine the genes which are regulated by melatonin concentrations. 

Minor

How many mares were aspirated each cycle?  How many follicles per mare?  Was cycle stage recorded or taken into account?  How did the presence of a CL affect gene expression or melatonin concentration?

How did you separate oocytes from different size follicles? 

Did you mature these oocytes?  Was there any difference in maturation rates and any other measured parameters? 

Line 33 – remove “an” 

Line 33 – change “monthly performed” to “performed monthly” – change throughout manuscript

Line 42-44 – Soften this statement

Author Response

As its main purpose, the paper seeks to evaluate seasonal variations in melatonin concentration in follicular fluid and mRNA expression of melatonin-related genes in mare developing follicles (small, medium and large). Given the results obtained, a correlation analysis was performed, as suggested by the reviewer.

We collected only follicular cells (granulosa and probably theca cells) for the samples. No cumulus-oocyte complexes were recovered.

The 2-ΔΔCt method, also called Livak method, is a relative quantification technique (described in Applied Biosystems User Bulletin #2). This quantification method is commonly used in our laboratory and has generated a lot of publications.

The samples were collected monthly and the number of mares and follicles per mare/month are shown in table 2.

The text was revised according to the suggestions of the reviewer.

Reviewer 2 Report

Reproduction seasonality in various domesticated animals represents an important limitation in assisted reproduction programs. Research is needed not only to enlighten the physiology of this incidence but also to examine and propose methodologies to reduce this phenomenon. Under this prism this study is of high importance. The manuscript is well written and the finding supported by the results but there are several parts that have to be improved before publication concerning the way they are presented, as follows: 

lines 33-35. The sentence is hard to understand, please rephrase. Probably “an” should be deleted in line 33

Although the introduction is well written providing a nice background and explanation of reproduction seasonality in mares, there is much information missing that has to be added. For instance in other mammals such as sheep, mutations in melatonin receptor gene have been associated with reproduction seasonality (Giantsis et al. 2016, 10.1186/s40709-016-0050-y, He et al. 2019, 10.1111/rda.13538). Is this also the same for mares?

Additionally some info regarding the mares should be added and their importance in the region of Brazil has to be added.

Materials and methods would be better divided in different sections

In mRNA quantification, several necessary parts are missing, conditions and volumes and more specifically which part of each gene is amplified with each primer pair

The conclusion regarding the potential use of melatonin in equine ART programs is a proposal of very high interest. It should be further analysed, probably with paradigms of other animals.

Author Response

Studies on melatonin receptors in horses are rare and there are no reports on melatonin receptor gene anomalies linked to reproductive seasonality in mares.

The information about accession number in National center of Biotechnology and primer sequences are listed in table 1.

The text was revised according to the suggestions of the reviewer.

Round 2

Reviewer 2 Report

I am ok with the revised version